# Burden of Disease (BoD) Assessment to Estimate Risk Factors Impact in a Real Nanomanufacturing Scenario

**DOI:** 10.3390/nano12224089

**Published:** 2022-11-21

**Authors:** Antti Joonas Koivisto, Marko Altin, Irini Furxhi, Maxime Eliat, Sara Trabucco, Magda Blosi, Jesús Lopez de Ipiña, Franco Belosi, Anna Costa

**Affiliations:** 1Air Pollution Management APM, Mattilanmäki 38, 33610 Tampere, Finland; 2Institute for Atmospheric and Earth System Research (INAR), University of Helsinki, PL 64, 00014 Helsinki, Finland; 3ARCHE Consulting, Liefkensstraat 35D, 9032 Wondelgem, Belgium; 4Witek s.r.l., Via Siena 47, 50142 Firenze, Italy; 5Transgero Limited, Cullinagh, Newcastle West, Co. Limerick, V42 V384 Limerick, Ireland; 6Department of Accounting and Finance, Kemmy Business School, University of Limerick, V94 T9PX Limerick, Ireland; 7CNR-ISAC, Institute of Atmospheric Sciences and Climate, National Research Council of Italy, Via Gobetti, 101, 40129 Bologna, Italy; 8ISTEC-CNR, Institute of Science and Technology for Ceramics, CNR, National Research Council, Via Granarolo 64, 48018 Faenza, Italy; 9TECNALIA, Basque Research and Technology Alliance (BRTA), Parque Tecnológico de Alava, Leonardo Da Vinci 11, 01510 Miñano, Spain

**Keywords:** burden of disease (BoD), disability adjusted life year (DALY), spray coating, emission, gaussian plume model, soil accumulation, risk assessment, inhalation exposure, nanomaterial, safety

## Abstract

An industrial nanocoating process air emissions impact on public health was quantified by using the burden of disease (BoD) concept. The health loss was calculated in Disability Adjusted Life Years (DALYs), which is an absolute metric that enables comparisons of the health impacts of different causes. Here, the health loss was compared with generally accepted risk levels for air pollution. Exposure response functions were not available for Ag nanoform. The health loss for TiO_2_ nanoform emissions were 0.0006 DALYs per 100,000 persons per year. Moreover, the exposure risk characterization was performed by comparing the ground level air concentrations with framework values. The exposure levels were ca. 3 and 18 times lower than the derived limit values of 0.1 μg-TiO_2_/m^3^ and 0.01 μg-Ag/m^3^ for the general population. The accumulations of TiO_2_ and Ag nanoforms on the soil top layer were estimated to be up to 85 μg-TiO_2_/kg and 1.4 μg-Ag/kg which was considered low as compared to measured elemental TiO_2_ and Ag concentrations. This assessment reveals that the spray coating process air emissions are adequately controlled. This study demonstrated how the BoD concept can be applied to quantify health impacts of nanoform outdoor air emissions from an industrial site.

## 1. Introduction

The control of industrial air emissions in Europe, is based on (i) source-specific emission standards, (ii) national emission reduction targets, established in the national emission ceilings, and (iii) ambient air quality directives, including air pollution limit values [1]. For nanomaterials, there are not yet specific standards or directives for industrial air emissions or population level exposure limit values. This is a challenge for industry manufacturing or processing nanoforms to evaluate if the air emissions are adequately controlled and does not cause an excessive risk for population health. Here it is demonstrated how a Burden of disease (BoD) concept can be used to evaluate health impacts by calculating population attributable exposure and exposure response functions for nanoforms. Then, the health impacts by nanoform emissions can be compared with well-accepted risk levels established for the general population exposed to air pollution [2,3]. This allows to evaluate if the risk level by nanoform emissions can be considered acceptable and adequately controlled.

The BoD assessment is used to estimate risk factors impact on population health loss in a systematic manner. The approach has been promoted to improve the current European Union-Health Information System (EU-HIS) [4]. BoD assessment considers how much is emitted, how are the people exposed to emissions, and what are the pollution components effects on human health. The overall assessment evaluates the impact of the risk factor in Disability Adjusted Life Years (DALYs), which is a health gap metric between the current and an ideal health situation of the population. DALY is an absolute metric particularly encompassing since it is comparable with other morbidity or mortality factors, such as smoking, diet, hygiene, or low physical activity and combines the estimation of time lived with disability and time lost due to premature mortality. This allows the estimations of the benefits of an ideal zero emission objective compared to the current situation and how emission impact relates to the currently prevailing situation. The BoD method is applied by the World Health Organization [5,6] and is used in Europe [7] for the health and environmental health impact assessment [8].

Some studies regarding the burden of disease of nanoforms exist in the literature, for example Yang et al. (2019) [9], utilized a compartmentalized physiologically based alveolar deposition model to estimate the lung disease burden posed by airborne Ag nanoforms emitted by consumer spray products. Forest et al. (2021) [10] investigated the relationship between occupational exposure, lung burden, and lung disease of nanoforms particles released unintentionally. However, the majority of the literature is focused on the potential beneficial usage of nanoforms, such as for antimicrobial agents [11] or improving existing treatment modalities [12,13]. DALYs have been used frequently for ambient air pollution purposes of fine and coarse particulate matter [14]. In addition, few studies exist for the assessment of indoor exposure, for example Walser et al. (2015) [15] presented a framework which goes beyond traditional LCA including nanospecific fate parameters as well as guidance on the development of effect and characterization factors for inhaled nanoforms.

In this study, the BoD methodology was used to evaluate the Witek’s spray coating process impact on a population level exposure and health. Spray-coating is a widely used industrial technique to deposit a wide variety of different shaped nanoparticles (NPs) on different substrates [16]. Witek uses the spray coating process to produce self-cleaning/self-purifying polyester and plastic surfaces. The coating is performed utilizing titanium dioxide nanoforms doped with nitrogen (TiO_2_N) and silver nanoforms capped by hydroxyethylcellulose (AgHEC).

The air emissions of the Witek process, characterized in a previous study [17], focused on occupational exposure and setting safe conditions of use for the spray process. This study applies the emission rates to evaluate the risk related to general population inhalation exposure and nanoforms soil accumulation. The risk assessment is used to decide if the factory outdoor air emissions are adequately controlled in the current situation and in case of scaling up the production under a reasonable worst-case emission (RWC) scenario. The risk characterization is performed by comparing the ground level air concentrations with health-based framework values derived for the general population from proposed occupational exposure limits for nanosized TiO_2_ and Ag. To our current knowledge, such a systemic approach is innovative in the field of nano-manufacturing processes.

## 2. Materials and Methods

The spray coating machine is conveyor belt operated and designed for coating up to 120 cm wide textile and plastic substrates at speeds ranging from 0.1 to 1 m/min. The coating machine (length 22 m, width 3.5 m and height 3 m) is described in detail by Del Secco et al. (2022) [18] and Koivisto et al. (2022) [17]. The coating process is performed by operating 1 to 4 air spray guns, each operating at a constant coating suspension flow rate of 200 mL/min. Coating suspensions are ethanol based containing 1 wt.% TiO_2_-N and water based containing 0.01, 0.05, or 0.1 wt.% Ag-HEC applied on polymethyl methacrylate (PMMA) or textile substrates.

The spraying is performed inside a ventilated chamber where overspray particles are ventilated via local exhaust ventilation (LEV) to outdoor air via a chimney. The chamber was mechanically ventilated with filtered air extracted from the room at a flow rate of 48.6 m^3^/min. The chamber temperature and relative humidity is specified by the room conditions. The LEV was equipped with a M5 type filter whose filtration efficiency for 0.4 μm particles is between 40% and 60% (EN 779:2012; EN 1822:2019) [19,20]. The M5 filter filtration efficiency was measured by using a non-standard method in field and laboratory (Appendix A). A fraction of overspray particles is released from the chamber openings to the work area where particles are expected to be removed mainly by the general ventilation to the outdoor air without filtration.

### 2.1. Air Emissions under Highest Reasonable Production Scenario

Koivisto et al. (2022) [17] characterized the atomized coating suspension mass flows to the substrate, LEV, and work room finding that the substrate, suspension type (TiO_2_ or AgHEC), number of spray guns, and AgHEC concentration had a significant effect on the transfer efficiency (a fraction that is deposited to the substrate; Tan and Flynn, 2002 [21]) and emission factors. The highest emission factors were selected for this study.

The air emissions from the coating process are calculated for RWC conditions, i.e., for the highest production volumes and material usage (Table 1). Process parameters were selected to favor the highest emissions in a full capacity production scenario where the coating machine is assumed to operate 5 days per week through a year (260 d/year) and 8 h per day (124,800 min per year). The process emissions were assumed to occur between 8:00 and 16:00. This corresponds to an annual nanoform consumption of 823.7 kg TiO_2_N and 49.9 kg AgHEC.

The LEV exhaust chimney height is 13 m with an inner diameter of 0.6 m where the overspray nanoforms are released to the outdoor air. The exhaust air temperature varies according to the room temperature, which is assumed to be 23 °C in the ambient air dispersion modeling. The emission rates from LEV to outdoors for TiO_2_N and AgHEC coating processes are 150 mg-TiO_2_/min and 2.8 mg-Ag/min, respectively. Fugitive emissions are assumed to be released through the building envelope without losses, which results in fugitive emission rates of 15.1 mg-TiO_2_/min and 0.012 mg-Ag/min. For fugitive emissions dispersion modeling, the building is simplified as a point source at a height of 3 m, diameter of 10 m, and volume flow of 5400 m^3^/h.

### 2.2. Air Emissions Dispersion Model

A Gaussian plume model estimates the transport of the airborne pollutants from a point source by diffusion (due to turbulent eddy motion), advection (due to the wind), and removal by precipitation [22]. In this study we used the bi-Gaussian plume model IMPACT (Immission Prognosis Air Concentration Tool) to calculate the industrial, residential, traffic, and agricultural emissions impact on the ground level air concentrations and depositions on a local scale [23]. IMPACT and its predecessor IDFM (Immission Frequency Distribution Model) have been tested and validated in numerous studies [24,25,26,27,28]. The Gaussian plume model is a generally accepted method to calculate dispersion and deposition of pollutants from various sources (e.g., [22]).

The parameters used in the dispersion model include weather conditions (wind speed and direction, atmospheric stability, and ambient air temperature) and source parameters (source location and height, source type, gas exit velocity, exit temperature and chemical species mass flow rate). The model assumes constant meteorology in time and space. This limits the model use on hilly terrain, long distance predictions, and time resolution. The model predictability depends strongly on the emission source characterization, meteorological data representativeness, and pollution deposition velocities [29]. Meteorological data specify the plume raise behavior and direction, the atmospheric stability class, i.e., width of the gaussian plume shape, and the wet deposition. The IMPACT model parametrization is fully described in the user manual [23].

Meteorological measurement data sets are incorporated into IMPACT for default dispersion modeling. Here was used meteorological data measured in Luchtbal, Antwerp, between 1 January 2007 and 31 December 2011 at a height of 24 m. Plume rise, Δ*h* (m), from stack exit is inversionally proportional to wind speed as $\Delta h\propto 1/u_{h_{g}}$, where $u_{h_{g}}$ (m/s) is the wind speed at the source height $h_{g}$ (m). A stack tip downwash occurs when the stack exit flow velocity, *v* (m/s), is low compared to the wind speed at the source height, $u_{h_{g}}$ (m/s). The stack tip downwash is corrected when $v/u_{h_{g}}<1.5$. The stack emissions depend only on the process and are independent of meteorological conditions.

Dry deposition velocity can be estimated by using a surface layer resistance model [29]. The deposition velocity depends on specific properties of the particles, of the atmospheric structure, and of the deposition surface. Currently, there is no single accepted theoretical description of the dry deposition phenomena because of the deposition processes’ complexity and the lack of experimental data covering all scenarios. Geometric mean diameter of TiO_2_N and AgHEC overspray agglomerated particles were in the range of 1 to 2 μm [17]. The deposition velocity for that particle size range varies from ca. 0.005 to 9 cm/s depending on the surface [29]. We selected a geometric mean deposition velocity of 0.16 cm/s to estimate the dry deposition velocity. This may underestimate the deposition in areas with high buildings or surrounded by forests or hills.

Wet deposition occurs at the time of precipitation and is relevant mainly for gaseous substances with high solubility (>1000 mg/L). Modeling a wet deposition using a washout coefficient means that all the physical and chemical processes related to the pollutant removal are represented by one factor. Washout coefficient is dependent on a large number of parameters, such as rainfall rate, rain droplet size, and aerosol size distributions and the concentrations [30]. Here, a generally accepted conservative value of 0.0004 1/s is used to describe the wet deposition of highly soluble and volatile substances [23,30].

The receptor grid was set as 2 × 2 km where the factory was placed in the center of the map. The number of receptor points was 400 distributed equally within the modelling area. A standard 1.5 m receptor height is used to calculate ground level concentrations. This represents human inhalation exposure.

### 2.3. Emission Impact Assessment as Disability Adjusted Life Years (DALYs)

The environmental BoD method provides the loss of healthy life-years due to ill health or premature death caused by an environmental risk factor on a population level. BoD method combines morbidity (Years Lived with Disability, YLD) and mortality (Years of Life Lost, YLL) into one comparable unit called Disability Adjusted Life Years (DALYs), i.e., DALY = YLD + YLL. DALYs are calculated multiplying the incidence and duration or prevalence of the health condition with the disability weight which reflects the severity of the disease and is scaled from 0 (perfect health) to 1 (equivalent to death) (1 DALY = 1 YLL). Hänninen and Knol (2011) [31] describe four methods for calculating the disease burden for relative risk functions and for unit risk functions.

Saber et al. (2019) [32] calculated the unit risk of 2.1 extra lung cancer cases per 100,000 exposed persons at an exposure level of 1 μg/m^3^ TiO_2_ nanoforms in air. Response functions for human inhalation exposure to Ag fumes, nanoforms or fine dusts are not available [33,34]. Because of limited data availability, the health burden of the coating process emissions was calculated only for TiO_2_ nanoforms emissions.

The DALYs due to lung cancer caused by TiO_2_ nanoforms were calculated with an attributable incidence (number of new cases per year) AI, and by converting the number of new cancer cases per year to DALYs per year by using following equations [31,35]:(1)$$\mathrm{AI}=E\times {\mathrm{UR}}_{\mathrm{TiO}2}\times \mathrm{POP}$$
(2)$$\mathrm{DALY}=\mathrm{AI}/\mathrm{ALE}\times {\mathrm{DALY}}_{\mathrm{lung}\;\mathrm{cancer}}$$
where E: exposure to TiO_2_ nanoforms (μg/m^3^), UR_TiO2_: a lifetime lung cancer risk (1/μg/m^3^), POP: exposed population. The lifetime cancer risk of the population was divided with the average life expectancy, ALE (years), to estimate new cases of cancers per year. The number of new cancer cases per year was converted to DALYs by multiplying the number of cases with the DALY loss of one lung cancer, DALYlung cancers (DALY/cancer). The parametrization is given in Table 2.

### 2.4. Emission Impact Assessment as Disability Adjusted Life Years (DALYs)

There are no legally binding inhalation exposure limit values for nanoforms. ECHA (2022) [39] reports a derived no effect level (DNEL) 2 μg/m^3^ for nanoAg for general population inhalation exposure. Here, the exposure was compared with recommended exposure limit values specified for 8 h time weighted average occupational exposure which were extrapolated for the general population as 24 -h continuous exposure.

Proposed OEL values vary for TiO_2_ nanoforms from 0.8 to 5000 μg-TiO_2_/m^3^ and for Ag nanoforms from 0.098 to 10 μg-Ag/m^3^ when given in different size fractions and specified under different experimental conditions [40,41,42]. We assume that OEL value can be extrapolated for the general population by scaling 40 h work week exposure to continuous weekly exposure by a factor of 4 and applying an assessment factor of 2 to describe children and senior sensitivity [43]. The lower range of the proposed OELs by using assignment factor of 8 results in general population limits of 0.1 μg-TiO_2_/m^3^ and 0.01 μg-Ag/m^3^. These limit values are indicative framework values used in this study.

### 2.5. Accumulation on Top Soil

Soil accumulation was estimated according to ECHA Chapter R.16, A.16-3.3.6 “Calculation of PEClocal for the soil compartment”. This is a single compartment model where the surface concentration is described with the deposition flux and a first-order rate constant (Equation R.16-39 in ECHA Chapter R.16):(3)$$\frac{dC_{soil}}{dt}=-k\cdot C_{soil}+D_{air}\;,$$
where *D_air_* (μg/m^2^/day) is the peak deposition flux, *t* (day) the time, *k* (1/day) the first-order rate constant for removal from top soil, and *C_soil_* (μg/kg) the top soil concentration. The model assumes that the top soil concentration is fully mixed all the time and there is no mass transfer between other surface compartments, i.e., the pollutant enters the soil top layer from air and is removed from bottom to deeper into the ground. The flows are irreversible, which means that the pollutant cannot return back after it is removed from the top layer. Here, the model was simplified by assuming that the pollutant cannot evaporate from the top layer and there is no degradation or other losses via biota. Thus, the only removal pathway is leaching by rainwater. The top layer was parametrized by using default parameters by ECHA: soil density 1700 kg/m^3^ ww (wet weight; dry weight (dw) is 1500 kg/m^3^), surface thickness is 0.2 m, and the initial TiO_2_ and Ag nanoform concentrations are 0 μg/kg.

The conservative estimates for two nano-TiO_2_ soil–water partition coefficient, *K_d_* (L/kg), were < 495 L/kg (UV Titan M262) and < 95 L/kg (P25) [44,45]. Higher *K_d_* value favors higher accumulation which was selected to estimate the soil accumulation upper limit under steady state. For Ag nanoparticle soil–water partition coefficient is up to 10,000 L/kg depending on the Ag nanoforms type and soil [46,47], which was used here as the precautionary value to estimate the Ag nanoparticle accumulation on soil top layer.

The loss rate leaching was calculated according to ECHA Chapter R.16, A.16-3.2 “Fate and distribution in environment”. The soil fraction of air, water, and solids were set by following the ECHA default fractions as 0.2, 0.2, and 0.6, respectively. The air–water partitioning was assumed to be insignificant. According to ECHA Equation R.16-7, the *K_soil-water_* (m^3^/m^3^) for TiO_2_ and Ag nanoforms are 743 and 15 000 m^3^/m^3^, respectively. The first-order rate constant for leaching was calculated according to ECHA Equation R.16-46 by using the default fraction of rain water that infiltrates into soil of 0.25, rate of wet precipitation of 700 mm/year (0.00192 m/day), and a mixing depth of soil for grassland of 0.2 m. For TiO_2_ and Ag nanoforms, this results in first-order rate constants 3.2 × 10^−6^ and 1.6 × 10^−7^ 1/day, respectively.

## 3. Results

### 3.1. Ground Level Concentrations and Deposition Fluxes

Dispersion of the stack emissions was calculated without M5 filter efficiency. The M5 filter filtration efficiency was verified by measuring the spray chamber concentrations and LEV concentrations after the M5 filter (Appendix A). The measurements showed that the filter efficiency complied with the M5 filter classification.

To cover seasonal variation in meteorology, the ground level concentrations and deposition fluxes (accumulation) are given as annual averages. Simulation file and the report generated by the IMPACT model are available at Appendix A. The annual average ground level concentrations and deposition fluxes are shown for TiO_2_ and Ag coating scenarios in Figure 1 and Figure 2, respectively.

Average ground level concentrations and deposition fluxes for the 4 km^2^ model area are 1.08 ng-TiO_2_/m^3^ and 0.283 μg-TiO_2_/m^2^/day for TiO_2_ and 0.018 ng-Ag/m^3^ and 0.005 μg-Ag/m^2^/day for Ag. The annual average deposition was 413 g-TiO_2_ which corresponds to a 2% of the annual fugitive and chimney emissions, i.e., 98% of the factory emissions are transported beyond the modeling area. Respectively, for Ag emissions the annual average deposition was 7 g-Ag which corresponds to a 4% of the annual fugitive and chimney emissions.

### 3.2. Health Burden of Nano-TiO_2_ Particles

The attributable incidence is calculated by using the average concentrations and the sum of the exposed population in the modeling area since the unit risk is a linear function. The population in the model area was assumed to be 40,000 habitants (Table 2). The attributable incidence is 0.0009 cancers per year, i.e., one cancer incidence in 1100 years, which corresponds to 0.00024 DALYs per year. For Ag, it was not feasible to estimate DALYs due to the lack of exposure response functions.

Ground level peak concentrations given as annual averages were 36.8 ng-TiO_2_/m^3^ and 0.57 ng-Ag/m^3^, which are ca. 3 and 18 times lower than the limit values of 0.1 μg-TiO_2_/m^3^ and 0.01 μg-Ag/m^3^ derived from the proposed OELs for nanosized TiO_2_ and Ag.

### 3.3. TiO_2_ and Ag Nanoparticle Accumulation on the Soil Top Layer

The soil top layer concentrations were calculated by using the nano-TiO_2_ and nano-Ag peak deposition fluxes of 8.0 and 0.13 μg/m^2^/day and assuming a 10-year production time. According to ECHA Chapter R.16 Equation R.16-39 the soil top layer concentration in this production scenario would be 85 μg-TiO_2_/kg ww and 1.4 μg-Ag/kg ww.

## 4. Discussion

The objective of this study was to demonstrate how the concept and methodology of the burden of disease can be applied in the case of a nano-manufacturing spray coating process. The approach can be utilized for a well-justified regulatory health impact decision-making demonstrating how TiO_2_N and AgHEC emissions at ground level concentrations are comparable to proposed limit values derived from proposed OEL values, for the general population. The strength in this assessment is in alignment with well-accepted methodologies used by WHO and EEA in the context of air pollution impact assessment. The precautionary parametrization followed in this study allows also the utilization of results for a wide range of industrial emission scenarios with similar emission rates, expanding its applicability domain. For example, factories are rarely located in a city with a population density of 10,000 habitants per km^2^ and the emissions in this case were calculated by using the highest process emission rates and annual production volumes without an exhaust air filter considered.

The health burden by TiO_2_N air emissions was 0.0006 DALYs per 100,000 persons per year. The acceptable risk range is typically defined as one additional cancer in 1,000,000 persons per year [2,48]. The DALYs by Witek spraying process are low compared to DALYs caused by air pollution in high socio-demographic index (SDI) countries which varies from 80, e.g., Finland, Sweden, to 300, e.g., Belgium, Italy per 100,000 habitants [49]. The DALYs caused by the factory emissions in this simulated scenario would contribute only 0.075 × 10^−3^ % to 0.02 × 10^−3^ % to the air pollution DALY burden. The fraction would be significantly smaller for lower SDI countries, such as Serbia, Montenegro, Bulgaria, or Bosnia and Herzegovina, where air pollution is estimated to cause ca. 1500 DALYs per year. Low DALYs by Witek process is due to the low annual air emissions and low nano-TiO_2_ unit risk. TiO_2_ nanoforms unit risk for cancer is generally lower than those of metals, such as cobalt 3 × 10^−3^ 1/μg/m^3^ [50], nickel 0.24 × 10^−3^ 1/μg/m^3^ (95% confidence interval (P95) 0.011 × 10^−3^–0.24 × 10^−3^ 1/μg/m^3^) [51], hexavalent chromium 8.32 × 10^−3^ 1/μg/m^3^ (P95 3.59 × 10^−3^–17.4 × 10^−3^ 1/μg/m^3^) [52], arsenic 0.15 × 10^−3^ 1/μg/m^3^ [53], or cadmium 1.8 × 10^−3^ 1/μg/m^3^ [51]. Unit risk of antimony 0.00012 × 10^−3^ 1/μg/m^3^ as Sb trioxide [54] is lower than for TiO_2_. On the other hand, the risks associated with metals are significantly lower than fine particulate matter (PM_2.5_), nitrogen oxides (NO_x_), or ozone (O_3_) [55].

It can be concluded that the factory emissions impact of TiO_2_ nanoforms on general population health is insignificant compared to frequently found carcinogenic air toxics or hazardous air pollutants. It is more reasonable and efficient to improve the air quality impact by reducing e.g., traffic emissions or emissions from fossil fuel based energy production [56] than aiming to minimize or zero the air pollution in Witek spray coating process. This does not justify exempting use of energy-efficient emission controls but the balance between increased energy use and the health impact by energy production needs to be considered. TiO_2_N and AgHEC emissions ground level concentrations were below the limit values derived for the general population from proposed OEL values for nanosized TiO_2_ and Ag suggesting that emissions are adequately controlled even in the worst case scenarios. Based on this evaluation, Witek spray coating process ambient air emissions are adequately controlled.

All carcinogenic health effect factors are currently labelled as indicative since there is no consensus with respect to the human relevance of the overload mechanism of carcinogenicity observed in rats [57]. Deposition fluxes can be used to estimate the Witek process emissions on soil and water accumulation. Comparing the environmental impacts of surface coating processes is not an easy task, because of the number of parameters that can affect the final result [58]. However, we based this study on realistic exposure measurements and on process factors that had already been investigated by Koivisto et al. (2022) [14], making the selection of the factors scientifically sound, minimizing uncertainties as much as possible.

Environmental exposure models designed for nanoforms can be used to estimate the environmental accumulation [59]. Current challenges in environmental fate models for nanoforms is material characterization, such as transformation processes, and validation [59]. The interactions between nanoforms and exposure factors are complex. For example, recommended test protocols may have limited relevance compared to natural environments [60]. In this study the accumulation of TiO_2_ and Ag nanoforms were estimated by employing a single compartment model by using default soil parametrization given in ECHA R.16. This resulted in soil top layer concentrations of 85 μg-TiO^2^/kg ww and 1.4 μg-Ag/kg ww when using the peak deposition fluxes and 10-year production time. Nanoforms concentrations in soil are scarcely reported because of the similarity in the elemental composition of nanoforms and those of natural nanoforms and micron sized particles, e.g., pigments [61,62]. Baalousha et al., (2020) [63] measured TiO_2_ concentrations from 550 to 1800 mg/kg (dry or wet weight is not specified) in the storm green infrastructures soil top layer. The median Ti-concentration in top soils and sediments is around 6000 mg/kg (dry or wet weight is not specified) in European soils [61]. Solely Ag nanoparticle concentrations in soil are not reported but the median elemental Ag concentration in the soil is ca. 0.02 mg/kg [64].

## 5. Conclusions

Ground level concentrations and deposition fluxes were simulated for two manufacturing scenarios to estimate population exposure and environmental emissions in a local scale (4 km^2^) for TiO_2_ and Ag nanoforms under RWC manufacturing scenarios. Health burden was calculated for TiO_2_ manufacturing scenario in DALYs while for Ag it was not possible because exposure response functions were not available. Currently, there are no reference or limit values for the general population TiO_2_ or Ag nanoforms exposure. Thus, the proposed limit values for general population were derived from proposed OEL values for nanosized TiO_2_ and Ag.

It was estimated that the TiO_2_ emissions in RWC manufacturing scenario would result to 0.0006 DALYs per 100,000 persons per year. This is low compared to the impact of urban air pollution that causes from 80 to 1500 DALYs per 100,000 persons per year. TiO_2_N and AgHEC emissions ground level concentrations were ca. 3 and 18 times lower than the limit values of 0.1 μg-TiO_2_/m^3^ and 0.01 μg-Ag/m^3^ derived from the proposed OELs for nanosized TiO_2_ and Ag. The assessment shows that the air emissions from the Witek spray coating process are adequately controlled. TiO_2_ and Ag accumulation at the soil top layer concentration in the 10-year production scenario was estimated to be low as compared to respective elemental Ti and Ag concentrations in European soils.

This study demonstrates how to assess the safety of industrial nanomaterial emissions without applying air pollution standards for nanomaterials. The BoD concept can be used to evaluate the factory air emissions annual health burden that can be compared with the generally accepted risk levels. This helps to decide if the emissions are adequately controlled, such as in the case of Witek spray process.

## Figures and Tables

**Figure 1 nanomaterials-12-04089-f001:**
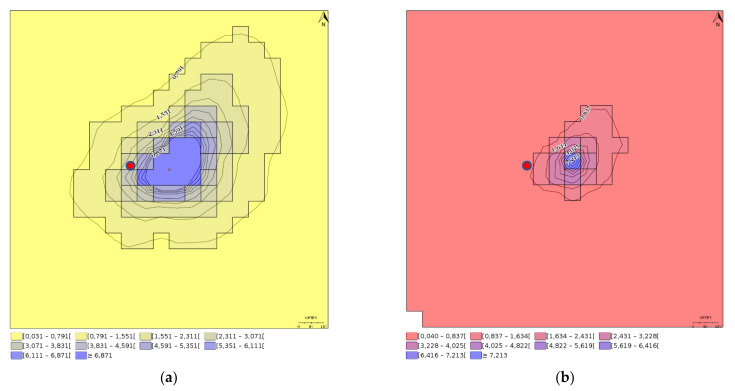
Annual average (**a**) ground level TiO_2_ concentrations in ng−TiO_2_/m^3^ and (**b**) deposition flux in μg−TiO_2_/m^2^/day. Red dot shows the chimney location. As an annual average, peak TiO_2_ ground level concentration is 36.8 ng−TiO_2_/m^3^ and peak deposition flux is 8.0 μg−TiO_2_/m^2^/day.

**Figure 2 nanomaterials-12-04089-f002:**
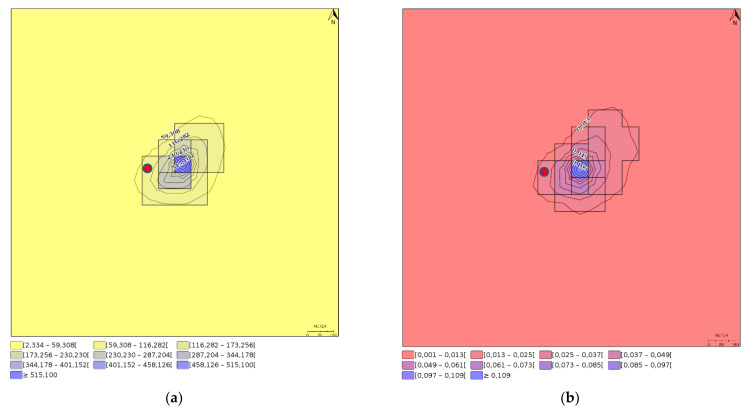
Annual average (**a**) ground level Ag concentrations in ng−Ag/m^3^ and (**b**) deposition flux in pg−Ag/m^2^/day. Red dot shows the chimney location. As an annual average, peak Ag ground level concentration is 570 ng−Ag/m^3^ and peak deposition flux is 0.13 μg−Ag/m^2^/day.

**Table 1 nanomaterials-12-04089-t001:** Spray process properties, transfer efficiencies and emission factors for TiO_2_N and AgHEC coating processes performed with four spray guns. TiO_2_N emission parameters are calculated as an average for textile and PMMA substrates and AgHEC emission parameters for textile substrates.

Property	TiO_2_N	AgHEC
Concentration, [wt.%]	1	0.1
NP mass flow per spray gun, [NP-g/min]	1.65	0.2
Number of spray guns	4	4
Transfer efficiency, [%]	99.5	99.63
EFroom, [NP-mg/NP-g]	2.288	0.029
EFLEV, [NP-mg/NP-g]	22.7	3.5
Annual production time, [min]	124,800	124,800
Mass of annually sprayed NPs, kg	823.7	49.9
Emission to room, [g]	1885	1.45
Emission to LEV (without the M5 filter), [g]	18,698	175

**Table 2 nanomaterials-12-04089-t002:** Parameters for TiO_2_ spray coating emissions DALY assessment.

Variable	Value	Description
Exposure	Varies (μg/m^3^/km^2^)	Exposure is based on spatial distribution of calculated ground level concentrations.
Health endpoint	Tracheal, bronchus and lung (TBL) cancer	For a lifetime lung cancer risk per μg-TiO_2_/m^3^ for inhalation exposure [32].
Risk function: Unit risk (UR)	0.021 × 10^−3^ cancer/μg/m^3^	
POP	10,000 1/km^2^	The coating factory is assumed to be located in a densely populated city [36]. Population density is assumed to be uniform.
Health loss per cancer incidence, DALYlung cancer	21.8 DALY/cancer	The number of new cancer cases per year was converted to DALYs by multiplying the number of TBL cancer cases with the mean DALY loss of one lung cancer. The DALY loss per cancer was estimated from the Global Burden of Disease 2019 Cancer Collabo-ration (2022) [37]. In 2019, TBL cancer were estimated to cause 45.9 million (95% uncertainty intervals (UI), 42.3–49.3 million) DALYs due to 2.26 million (95% UI, 2.07–2.45 million) incident TBL cases. Based on upper 95% UI and incident TBL cases, the DALYlung cancer is 21.8 DALY/cancer.
Average life expectancy	83.8 years	Average life expectancy in EU at birth for 2017–2019 [38].

## Data Availability

The data presented in this study are available in this article, Appendix A, and the references herein.

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
