# Peer review of "Burden of Disease (BoD) Assessment to Estimate Risk Factors Impact in a Real Nanomanufacturing Scenario"

_nanomaterials, 2022, doi:10.3390/nano12224089_

Round 1
Reviewer 1 Report
This is an interesting paper of emissions from nano-coating materials and impact on air quality, soil and health.
In the intorduction section the authors must explain clearly what was the motivation of this work and the context in which it took place. Moreover, they must explain even it is mentioned elsewhere what is the Witek's process and why it is important.
Section 2.1 should be made more elaborate to explain how this is emitted and measured. The atmospheric conditions inside the chamber such as temperature, relative humidity, maybe the range in which emission happens are important information that is missing.
In section 2.2 please explain how the stack height and outdoor meteorological conditions connect with the test chamber conditions described above and also please explain how and where nano materials exist and are emitted into the atmosphere. Which processes (industrial or other) are involved?
In section 3.1 what is the M5 filter? It was not mentioned in section 2.
Finally, the authors conclude that the nano materials studied have a very low impact on health based on the DALY's estimated. So what is the significance of this work? For instance a review on the existing nano-coating industrial units would be helpful to give a background on the subject. As well, other models such as lung deposition ones might be more interesting to use. Addition of official health reports by EEA, WHO and other organisations might explain the importance of this work.
Author Response
This is an interesting paper of emissions from nano-coating materials and impact on air quality, soil and health.
In the introduction section the authors must explain clearly what was the motivation of this work and the context in which it took place.
We agree. We added to the first section: ”For nanomaterial’s, there are not yet specific standards or directives for industrial air emissions or population level exposure limit values. This is a challenge for industry man-ufacturing or processing nanomaterials to evaluate if the air emissions are adequately controlled and does not cause an excessive risk for population health. Here is demon-strated how a Burden of disease (BoD) concept can be used to evaluate health impacts by calculating population attributable exposure and exposure response functions for nano-materials. Then, the health impacts by nanomaterial emissions can be compared with well-accepted risk levels established for general population exposed to air pollution (WHO, 2006, 2000). This allows to evaluate if the risk level by nanomaterial emissions can be considered acceptable and adequately controlled.”
Moreover, they must explain even it is mentioned elsewhere what is the Witek's process and why it is important.
We agree. We added to introduction that “Spray‐coating is widely used industrial technique to deposit a wide variety of different shaped nanoparticles (NPs) on different substrates (Bekker et al., 2014). Witek use the spray coating process to produce self‐cleaning/selfpurifying polyester and plastic surfaces.”
Section 2.1 should be made more elaborate to explain how this is emitted and measured. The atmospheric conditions inside the chamber such as temperature, relative humidity, maybe the range in which emission happens are important information that is missing.
We agree. We added to section 2 “The chamber was mechanically ventilated with filtered air extracted from the room at a flow rate of 48.6 m3/min. The chamber temperature and relative humidity is specified by the room conditions.” We also added to section 2.1 “The process emissions were assumed to occur between 8:00 – 16:00.”
In section 2.2 please explain how the stack height and outdoor meteorological conditions connect with the test chamber conditions described above and also please explain how and where nano materials exist and are emitted into the atmosphere. Which processes (industrial or other) are involved?
We agree. We added “The stack emissions depend only on the process and are independent of meteorological conditions.” In section 2.1 we clarified the where the nanoforms are released to outdoor air. Fugitive emissions are explained in the same section.
In section 3.1 what is the M5 filter? It was not mentioned in section 2.
This was mentioned in Section 2 “The LEV was equipped with a M5 type filter which filtration efficiency for 0.4 μm particles is between 40% and 60% (EN 779:2012; EN 1822:2019) [16,17]. The M5 filter filtration effi-ciency was measured by using a non-standard method in field and laboratory (Supplementary Material).”
Finally, the authors conclude that the nano materials studied have a very low impact on health based on the DALY's estimated. So what is the significance of this work? For instance a review on the existing nano-coating industrial units would be helpful to give a background on the subject. As well, other models such as lung deposition ones might be more interesting to use. Addition of official health reports by EEA, WHO and other organisations might explain the importance of this work.
In Discussion, we state that “The objective of this study was to demonstrate how the concept and methodology of the burden of disease can be applied in the case of a nano-manufacturing spray coating process.” We agree, that this approach should be applied more broadly, e.g. by covering the whole nano-TiO2 industry emissions to understand risks related the whole sector. Lung deposition, i.e. dose estimates, are needed if dose-responses are available. However, currently the health effects are estimated by using exposure-response functions.

Reviewer 2 Report
The authors reported the study of airborne exposure to the coating particles of TiO2 and Ag formers risk factor estimation to the health in real nanomanufacturing scenarios. The various factors were estimated for both TiO2 and Ag formers in inhalation and settlement in the soils. The manuscript shows exemplary data to prospect disability-adjusted life years. The manuscript is good and could be accepted for publication after minor revision.
1- the abstract was written generally, it is better to redesign the abstract with forcing on the merit of the figure.
2- It is better to reduce the conclusion part.
3- It is better to remove the conclusion from the supplementary materials.
Author Response
The authors reported the study of airborne exposure to the coating particles of TiO2 and Ag formers risk factor estimation to the health in real nanomanufacturing scenarios. The various factors were estimated for both TiO2 and Ag formers in inhalation and settlement in the soils. The manuscript shows exemplary data to prospect disability-adjusted life years. The manuscript is good and could be accepted for publication after minor revision.
1- the abstract was written generally, it is better to redesign the abstract with forcing on the merit of the figure.
The abstract was better focused on the application of BoD concept for air emissions health impact assessment in nano-industry.
2- It is better to reduce the conclusion part.
We agree. Conclusions section was condensed.
3- It is better to remove the conclusion from the supplementary materials.
We agree. Conclusions was removed

Round 2
Reviewer 1 Report
The article does not have serious flaws, but in my opinion it is of low interest. It would add value to include a short para in the conclusions section with the use and future perspectives of such work and of the wider use of the materials.
Author Response
We added a short section below to the conclusions and made a language proof reading.
“This study demonstrates how to assess the safety of industrial nanomaterial emissions without applying air pollution standards for nanomaterials. The BoD concept can be used to evaluate the factory air emissions annual health burden that can be compared with the generally accepted risk levels. This help to decide if the emissions are adequately controlled, such as in the case of Witek spray process.”
